# Cisplatin Resistance and Redox-Metabolic Vulnerability: A Second Alteration

**DOI:** 10.3390/ijms22147379

**Published:** 2021-07-09

**Authors:** Medhi Wangpaichitr, George Theodoropoulos, Dan J. M. Nguyen, Chunjing Wu, Sydney A. Spector, Lynn G. Feun, Niramol Savaraj

**Affiliations:** 1Department of Veterans Affairs, Miami VA Healthcare System, Research Service (151), Miami, FL 33125, USA; theogengr@yahoo.com (G.T.); j.nguyen3@miami.edu (D.J.M.N.); chunjing.wu@va.gov (C.W.); sydney.spector@va.gov (S.A.S.); 2Department of Surgery, Cardiothoracic Surgery, Miller School of Medicine, University of Miami, Miami, FL 33136, USA; 3Department of Medicine, Hematology/Oncology, Miller School of Medicine, University of Miami, Miami, FL 33136, USA; lfeun@med.miami.edu (L.G.F.); nsavaraj@med.miami.edu (N.S.); 4Department of Veterans Affairs, Miami VA Healthcare System, Hematology/Oncology, 1201 NW 16 Street, Room D1010, Miami, FL 33125, USA

**Keywords:** cisplatin resistance, metabolism, oxidative metabolism, reactive oxygen species

## Abstract

The development of drug resistance in tumors is a major obstacle to effective cancer chemotherapy and represents one of the most significant complications to improving long-term patient outcomes. Despite early positive responsiveness to platinum-based chemotherapy, the majority of lung cancer patients develop resistance. The development of a new combination therapy targeting cisplatin-resistant (CR) tumors may mark a major improvement as salvage therapy in these patients. The recent resurgence in research into cellular metabolism has again confirmed that cancer cells utilize aerobic glycolysis (“the Warburg effect”) to produce energy. Hence, this observation still remains a characteristic hallmark of altered metabolism in certain cancer cells. However, recent evidence promotes another concept wherein some tumors that acquire resistance to cisplatin undergo further metabolic alterations that increase tumor reliance on oxidative metabolism (OXMET) instead of glycolysis. Our review focuses on molecular changes that occur in tumors due to the relationship between metabolic demands and the importance of NAD^+^ in redox (ROS) metabolism and the crosstalk between PARP-1 (Poly (ADP ribose) polymerase-1) and SIRTs (sirtuins) in CR tumors. Finally, we discuss a role for the tumor metabolites of the kynurenine pathway (tryptophan catabolism) as effectors of immune cells in the tumor microenvironment during acquisition of resistance in CR cells. Understanding these concepts will form the basis for future targeting of CR cells by exploiting redox-metabolic changes and their consequences on immune cells in the tumor microenvironment as a new approach to improve overall therapeutic outcomes and survival in patients who fail cisplatin.

## 1. Introduction

Cisplatin is an active agent that is widely used in the treatment of several types of solid tumors and has been a gold standard in the treatment of many types of cancer, including lung (both small cell and non-small cell lung cancer), head and neck cancer, and ovarian cancer [1,2,3]. Cisplatin-induced toxicity involves various mechanisms, such as DNA adduct formation, mitochondrial dysfunction, and oxidative stress [4] The majority of patients will initially respond to cisplatin treatment; however, resistance to cisplatin is inevitable. The development of cisplatin resistance is complex and involves multiple mechanisms, which makes it difficult to overcome. Thus far, there are no available drugs that can reverse cisplatin resistance or selectively kill cisplatin-resistant (CR) cells.

It has become apparent that cancer cells rewire their metabolic reliance to glycolysis to promote growth, survival, and proliferation [5]. This is considered the “first” metabolic switch along their survival path. Nevertheless, emerging evidence implicates that CR cells undergo additional metabolic reprogramming wherein tumors become more reliant on oxidative metabolism (OXMET), resulting in higher accumulation of ROS [6,7,8]. Thus, alteration in redox status by modulation of ROS can influence cellular sensitivity to cisplatin.

The exact mechanism for a “second” metabolic switch from glycolysis to OXMET in the development of cisplatin-resistant tumor cells has not been elucidated. In this review, we summarize additional alterations in redox-metabolism that play a vital role in adaptive resistance to cisplatin and cellular metabolism. Furthermore, we focus on the features of tumor metabolism and redox as well as the kynurenine pathway (tryptophan catabolism) in relation to cisplatin resistance and propose how exploiting metabolic pathways could overcome drug resistance.

## 2. Cancer Cell and Metabolic Demand

Most if not all cancer cells are known to increase glycolysis demand to adapt with high proliferation rate and growth. Under aerobic glycolysis, cancer cells can rapidly produce building blocks such as ribonucleotides and amino acids as well as adenosine triphosphate (ATP). Because of rapid growth, this population of cancer cells are the target for standard chemotherapeutic agents such as cisplatin. More importantly, tumors utilize glycolysis instead of oxidative phosphorylation (OXPHOS) even with adequate supply of oxygen as their primary source of energy (“Warburg effect”) [9,10]. This effect may not be due to defective mitochondrial respiration but rather due to upregulation of glycolytic enzymes and glucose transporters [11,12]. However, there are also reports indicating that many types of cancer are indeed associated with mutations in TCA cycle enzymes [13,14,15]. Together, it is evident that the “first” metabolic switch from normal to cancer cell reprograms cells to glucose addiction; as a consequence, increased glucose uptake becomes one of the hallmarks for malignant transformation [16]. Due to this difference in energy metabolism between tumor and normal tissue, uptake of glucose analogs (fluoro-deoxy-glucose) has been utilized successfully in the development of a diagnostic imaging technique, fluoro-deoxy-glucose positron emission tomography (FDG-PET) for cancer detection [17]. However, PET-negative results (not taking up FDG) after chemotherapy does not always correlate with tumor response and can lead to detrimental false negative results. Based on these phenomena, we are firmly convinced that cancer cells undergo a “second” major metabolic rewiring when they acquire resistance to chemotherapy such as cisplatin.

### 2.1. Cisplatin Resistance and Glucose Metabolism

A straightforward scheme for cells to resist a cytotoxic drug is to not absorb it. Thus, decreased cisplatin accumulation is one of the most important mechanisms of resistance found in CR cells. Decreased hTCR1 (a copper transporter pump) expression, which transports cisplatin inside cells, has been shown to be one of the major contributory factors to decreased intracellular cisplatin accumulation [18,19]. Over-expression of hCTR1 signifies better response rates, better progression free survival, and overall survival among patients with stage III non-small-cell lung cancer [20]. Corresponding with this occurrence, it was recently shown that glucose consumption and uptake in CR cells is reduced in comparison with cisplatin-sensitive cells. The reduction in glucose consumption and uptake is due to the loss of glucose transporter-1 (GLUT1) expression on the plasma membrane of CR cells [21]. Hence, it is conceivable that the development of cisplatin resistance in cancer cells is the consequence of metabolic reprograming from glycolytic metabolism to oxidative metabolism (OXMET). By utilizing OXMET instead of glucose, CR cells slow down proliferation, which aids in escaping the effects of cisplatin, which is designed to target fast dividing cells. Importantly, metabolic changes in CR cells cause abnormal mitochondrial functions, such as electron leak, which in turn results in higher levels of mitochondrial reactive oxygen species (ROS) [21,22]. Increased ROS can facilitate the activation of receptor tyrosine kinase signaling as well as PI3K/AKT, which plays a vital role in cell growth/proliferation, survival, and motility [23,24] (Figure 1)**.**

Additionally, lowered expression of key glycolytic enzymes is reported in CR cells compared with cisplatin-sensitive cells. For instance, the expression of hexokinase-2 (HK2) and lactic dehydrogenase-A (LDHA) is reported to be significantly lower in cisplatin-resistant cells compared with sensitive cells [25]. This finding points to a concept where the glycolytic pathway is not as active in CR tumors compared with sensitive tumors. Furthermore, the decreased expression of LDHA coincides with decreased lactate production from CR cells compared with sensitive cells, also signaling that cisplatin-resistant cells no longer depend on glycolysis. We have found that cisplatin-treated and already-resistant cells (CR), expressing low levels of the final rate-limiting enzyme of the glycolytic pathway, PKM2, have survived treatment and consume more oxygen and produce less lactate, indicating a shift toward OXMET utilization [26]. In addition to our work, other studies have shown that PKM2 protein and activity levels were lower in CR human gastric carcinoma cell lines compared with their parental cell lines. Further suppression of PKM2 expression using antisense oligonucleotides increased cisplatin resistance in these CR cells [27]. In a clinical study, PKM2 mRNA levels were measured in tumors treated with oxaliplatin in combination with 5-fluorouracil, and the tumors with the lowest levels of PKM2 attained the lowest response rates to treatment [28]. In contrast to these findings, others showed that increased PKM2 levels have been linked with resistance to 5-fluorouracil in patients with colorectal cancer [29]. Moreover, eliminating expression of PKM2 in some naïve cells is reported to re-sensitize tumor cells to platinum-based therapies [30,31,32,33]. These studies hint that resistance mechanisms are tissue specific and resistance specific. Genetic silencing of PKM2 increased the efficacy of docetaxel- and cisplatin-mediated cell death in in vitro and lung cancer xenograft models [34,35]. These conflicting reports and the lack of a definitive role for PKM2 in the development of drug resistance have raised concerns about the potential of PKM2 as a valid cancer drug target. Nonetheless, it is irrefutable that CR cells rewire glycolytic metabolism for a survival advantage.

### 2.2. Cisplatin Resistance and Glutamine Anaplerosis

Since CR cells are not primarily reliant on glucose, they must be dependent on other carbon skeleton sources to survive. By utilizing OXMET, CR cells can metabolize glutamine to glutamate via the glutaminase (GLS) enzyme. Glutamate will then serve as an anaplerosis substrate for alpha keto-glutarate in the TCA cycle on which OXMET is dependent (Figure 2). With this modification in energy sourcing, it has been shown that CR cells consume more glutamine [6]. Importantly, CR cells have increased cellular energy or metabolic demand that outstrips the glutamine supply, making glutamine the conditionally essential amino acid for cell survival. We and others have shown that CR cell growth was very sensitive to glutamine deprivation, and confirmed that cells were unable to survive in glutamine-free media [6,36,37]. These finding were further supported when the exposure to the combination of glutaminase inhibitor (BPTES) and a platinum-based agent synergistically inhibited platinum-resistant ovarian cancers in vitro [36]. Hence, there are ample data at this time to support a dependence on glutamine in CR cells that rely on OXMET and the TCA cycle.

The higher consumption of glutamine served as an indicator that CR cells undergo a *“second”* metabolic reprograming. Some reports have shown that glutamine imported into cells is not totally utilized for anabolic metabolism [37]. Rather, a portion of glutamine/glutamate is shuttled out of the cell in exchange for amino acids inside the cell that directly activate mTOR (the mammalian target of rapamycin), which is a significant component of the PI3K family of protein kinases in cellular proliferation [38,39]. In fact, increased glutamate efflux is an important process for cells to generate glutathione (GSH) to cope with high intracellular ROS levels. This process is carried out through the x_c_^-^ system, a cystine–glutamate exchange transporter on the membrane composed of the xCT subunit, wherein glutamate is removed from the cell while cysteine is imported and later forms GSH (Figure 2) [40]. Cysteine is considered the rate-limiting precursor in this process. Sulfasalazine (SSZ), a known inhibitor of the x_c_^-^ system pump [41,42] had a synergistic effect when used in combination with cisplatin in colon cancer [43]. These investigators believed that cancers with increased expression of x_c_^-^ on the membrane may be more susceptible to cisplatin treatment. In addition, we also showed that blocking glutamate efflux by riluzole (an FDA-approved drug for amyotrophic lateral sclerosis and effective inhibitor of the xCT/cystine/glutamate pump) can selectively kill CR cells in vitro and in vivo [6]. In the treated cells, glutamate secretion was decreased since the cysteine antiport was inhibited, leading to elevated ROS and cell death without co-treatment with cisplatin. Moreover, riluzole can further decrease NAD+ (nicotinamide adenine dinucleotide) and LDHA expression, which in turn heightens oxidative stress in CR cells [6].

### 2.3. Cisplatin Resistance and Urea Cycle

Similar amino acid therapeutic approaches are being investigated for the treatment of arginine auxotrophic tumors. There are numerous reports of tumors such as melanoma, mesothelioma, prostate cancer, and hepatoma that are auxotrophic for arginine [44,45,46,47]. These tumors have low or no expression of argininosuccinate synthetase-1 (ASS1), which is a key enzyme in the urea cycle that catalyzes the synthesis of arginine from citrulline (Figure 2). Thus, the lack of ASS1 expression makes arginine an essential amino acid in these tumors.

Moreover, downregulation of ASS1 expression has been linked to poor prognosis with increased proliferation and invasion of cancer cells [48,49]. Methylation of the CpG islands within the ASS1 promoter has been reported many times as one of the mechanisms responsible for the loss of ASS1 expression in solid tumors [48,50,51]. Furthermore, patients treated with first-line platinum/paclitaxel for ovarian cancer had a poor overall and disease-free survival in tumors exhibiting methylated ASS1 compared with unmethylated ASS1 [50,52].

Interestingly, it has been shown that the epigenetic inactivation of ASS1 can be associated with selective resistance to platinum-based treatment in primary ovarian cancer-cultured cells [52]. Consistent with this finding, we also reported ASS1 silencing in ovarian cancer that acquired resistance to cisplatin [51]. ASS1 suppression is controlled by the transcriptional repressor HIF-1α, which occupies the E-box at the ASS1 promoter and blocks c-MYC from binding [51]. De-repression of ASS1 from HIF-1α binding allows c-Myc to activate ASS1 expression [53]. Tumors lacking the ASS1 enzyme require extracellular arginine in the circulation for survival [54,55,56]. Arginine deprivation therapy using the arginine degrading enzyme ADI-PEG20 (pegylated arginine deiminase) or human arginase 1 has been in various stages of clinical evaluation for targeting Arg-auxotrophic tumors [55]. ADI-PEG20 digests arginine into citrulline and ammonia [57], and human arginase 1 digests arginine into ornithine and urea [56]. ADI-PEG20 is safe, and the drug is only effective in patients whose tumors are negative for ASS protein expression. Together, these findings highlight the potential of arginine deprivation therapy as a strategy to overcome resistance to platinum-based drugs.

### 2.4. Cisplatin Resistance and Fatty Acids

In addition to amino acid metabolism having an increasing role in cisplatin resistance, emerging reports showed that upregulation of fatty acid synthesis is associated with poor prognosis [58,59,60] and that overexpression of fatty acid synthase (FASN; key regulator of the de novo synthesis of fatty acids) is correlated with markers of cellular proliferation [61] and interferes with drug efficacy [62]. Inhibiting FASN using orlistat can reverse acquired resistance to trastuzumab in breast and ovarian cancer cells [63,64], as well as hyper-sensitize breast cancer cells to doxorubicin, docetaxel, paclitaxel, or vinorelbine [65,66]. Others have shown that over-expression of FASN can be a predictive marker for cisplatin resistance and that inhibiting FASN can overcome cisplatin resistance in mice bearing breast cancer xenografts [67]. Moreover, combination treatment of C75 (FASN inhibitor) and cisplatin resulted in greater growth inhibition of ovarian cancer in vivo than cisplatin treatment alone [61]. Studies also suggested that FASN inhibition may work beyond suppression of FASN activity. It has been shown that FASN reduced both the expression of multidrug resistance protein (MDR) and multidrug resistance-associated protein (MRP-1) types of ABC membrane transporters that are practically ubiquitously expressed in tissues and found to pump chemotherapeutic agents out of cells [68,69]. Consistent with these findings, we also reported that CR lung cancer cells that over-expressed homolog MRP-4 [70] also possessed 15-fold more acetyl-CoA carboxylase (ACC; first enzyme in the committed step of fatty acid synthesis), and 5-fold more FASN when compared with parental cell counterparts [22]. Thus, overexpressing all the previously mentioned molecules that contribute to drug resistance and survival in the scheme described herein involves fatty acid synthesis. We also reported that TOFA (an allosteric inhibitor of ACC that blocks the synthesis of malonyl CoA) or C75 (FASN inhibitor) induced significant cell death in CR cells [22]. Overall, the data illustrated that modulation of fatty acid metabolism is an acquired resistance mechanism whose inhibition may represent a novel strategy for the treatment of tumors that are resistant to cisplatin.

## 3. Cancer Cell and Redox Balance

One of the mechanisms of cisplatin resistance development is connected to the influence of ROS-induced damage to DNA and cellular molecules. Under normal physiological conditions, ROS is tightly regulated by the balance between its production (oxidant) and both enzymatic and non-enzymatic elimination (antioxidant). To avoid oxidative damage, ROS is detoxified by catalase and two other systems: glutathione (GSH) and thioredoxin (TRX) [71,72,73]. In these systems, NADPH is required for regeneration of GSH and TRX through glutathione reductase and thioredoxin reductase, respectively [74,75]. NADPH also reactivates catalase when catalase is inactivated by H_2_O_2_ [76]. Contrary to the role of NADPH, studies have shown that NADPH could also significantly contribute to generation of oxidative stress through the activity of NADPH oxidase [77,78]. NADPH oxidase (NOX) is an enzyme that catalyzes the generation of intracellular superoxide from oxygen and NADPH (Figure 3). Increasing evidence has indicated that NOX activity is present not only in phagocytes but also in various tissues and cell types [77,79]. Importantly, NOX expressions have been involved with cisplatin resistance [80]. Therefore, it is clear that the alteration in redox status via ROS modulation can influence cellular sensitivity to cisplatin.

### 3.1. Cisplatin Resistance and GSH/TRX Antioxidant Systems

Numerous reports have shown that cisplatin targets thioredoxin reductase (TrxR), leading to increased intracellular ROS and resulting in growth arrest and subsequent cell death [81,82]. In order to adapt and survive at higher ROS levels and to avoid cell death caused by cisplatin, CR cells use less TRX and employ other antioxidant systems to compensate for the lack of TRX [83,84,85]. In fact, many have reported that CR cells have higher level of glutathione (GSH) proteins [86,87]. Lower intracellular TRX is due to protein degradation caused by cathepsin-D [88] or can be a consequence of increased TRX secretion, as we have reported [22]. TRX can be secreted via a special secretory pathway called the “leaderless pathway”, which is known to secrete low molecular weight proteins that lack a signal peptide [89,90,91]. The mechanism of how this pathway functions remains poorly understood. Nevertheless, increased TRX secretion usually occurs when cells are under stress [92,93,94] and is detected in patients who received cisplatin treatment [95,96]. It is very likely that CR cells secrete higher amounts of TRX due to continuous cellular stress, which in turn results in lower intracellular accumulation of TRX and consequently increased ROS. Interestingly, studies showed that using TRX inhibitor (PX-12) can lower the elevated levels of plasma TRX in cancer patients, which can be used as surrogate indicator for the inhibition of tumor growth or proliferation [97]. More importantly, it is possible that augmented TRX levels can be reflected in serum samples prior to tumor progression by the RECIST criteria and hence may be used as early markers for disease progression.

As for the role of GSH in CR cells, it has been shown that pre-incubation with the GSH inhibitor buthionine sulfoximine (BSO) induces massive cell death, whereas N-acetyl cysteine, a precursor of glutathione synthesis, improves the resistance to cisplatin treatment [98]. However, targeting GSH should be done with caution, since a decade ago, depletion of GSH was not successful in increasing the antitumor effects of cisplatin in clinic due to the fact that GSH can also bind to platinum, which further complicates treatment [99]. It is noteworthy that chemotherapeutic agents generate ROS as one of the potent mechanisms to eradicate tumor cells; therefore, it is not surprising that targeting ROS by antioxidants has yielded mixed results in the therapeutic efficacy of chemotherapy [100].

### 3.2. Cisplatin Resistance and NAD+

Nicotinamide adenine dinucleotide (NAD+) is a crucial co-enzyme of all organisms’ redox systems and is involved in many signaling pathways including cellular metabolism and DNA repair. In tumor cells, LDHA converts pyruvate to lactate and thereby generating NAD+, which will subsequently be reduced to NADH and used in glycolysis again (Figure 2). Thus, anaerobic glycolysis in cancer cells is dependent on NAD+. Moreover, NAD+ is also used as the precursor to generate NADP+ via NAD kinase (NADK) [101]. In its reduced form (NADPH), this molecule maintains and regenerates cellular detoxifying and antioxidant oxidative defense systems as mentioned. As a result, it is undeniable that the level of NAD+ can influence cell homeostasis and survival. Significant alterations in NAD+ levels have been reported in renal tissue treated with cisplatin. In this study, they showed that cisplatin treatment resulted in a decrease of NAD+ in renal tissue without significant changes of NADH level [102], thus suggesting that the decrease in the NAD+/NADH ratio by cisplatin is mainly caused by reduction of NAD+ levels. Therefore, it is possible that when cells became resistant to cisplatin, they adapted to survive under diminishing levels of NAD+. Our own findings that NAD+ levels are lower in CR cells are supported by other reports [6,103] and support the notion that CR cells undergo “*second”* metabolic reprograming and do not primarily depend on glycolysis.

The canonical role of NADPH oxidase (NOX) is to transport electrons across the plasma membrane, and in turn, to generate superoxide. The NOX family (i.e., NOX1, NOX4, etc.) is active in many important biological processes, including host defense, redox signaling, gene regulation, and post translation modifications [104,105]. Consumption of NAD+ by NOX in various cancer cells may also attenuate regeneration of TRX and GSH, thereby initiating an accumulation of intracellular ROS [106] (Figure 3). In addition, NOX may interfere with electron transport and affect ATP synthesis in mitochondria via activation of PKCε, mitoKATP, or modulation of mitochondria–thioredoxin activity [107]. Hence, it is possible that increased NOX can lead to *“second”* metabolic alterations that in turn lead to cisplatin resistance. In fact, it has been shown that NOX4 contributes to cisplatin resistance in renal cancer cells by modulation of pro-apoptotic and anti-apoptotic signaling, suggesting that NOX4 inhibition might enhance the efficacy of conventional cytotoxic drugs against renal cancers [80]. Additional support for a role in CR cell survival and tumor growth has been reported by others, who identified NOX4 as a critical downstream effector of inositol-trisphosphate 3-kinase B (ITP3K), a proliferative signal [108]. Therefore, targeting NOX4 could be used as an additional therapeutic agent to overcome resistance to cisplatin in future treatment.

## 4. Cancer Cells and NAD+/PARP-1/SIRTs Axis

Poly (ADP-ribose) polymerase 1 (PARP-1) and sirtuin (SIRT) are NAD+ dependent enzymes, and their interactions intertwine with cellular metabolism as well as the oxidative stress response. PARP-1 polymerizes ADP-ribose from NAD+ to the target protein, resulting in the formation of poly (ADP-ribose) or PAR, thus making PARP one of the major consumers of NAD+ [109,110]. Because PAR is negatively charged and noncovalently couples with nuclear proteins, PAR can act as a scaffolding for chromatin remodeling and DNA repair processes [111]. As for DNA damage repair, PARP-1 itself is its own target protein and is subjected to self-ribosylation (self-PAR) [112]. The synthesis and accumulation of a PAR chain then results in the recruitment of DNA repair scaffold enzymes (Figure 4, left panel). Ultimately, the activity of DNA ligase repairs DNA breaks. SIRTs also utilize NAD+ as a cofactor for their enzymatic activity. SIRTs have the dual role of protein modifications via deacetylation and ribosylation (Figure 4, right panel), which can lead to direct activation or inhibition of target transcriptional regulators as well as to the modification of their interaction profiles.

The interaction between PARP-1 and SIRT is exclusively regulated by cellular NAD+ levels. It has been known that the decrease of NAD+-enhanced PARP activity correlates with a downregulation of SIRT activity [113]. Similarly, the activation of SIRT reduces PARP activity [114]. Thus, the pharmacological intervention to modulate PARP-1 and SIRT enzymes is possible but requires further understanding in CR tumors because CP induces DNA damage.

### 4.1. Cisplatin Resistance and PARP-1

Poly (ADP-ribose) (PAR) and the PAR polymerase-1 (PARP-1) play a key role in maintaining DNA integrity in the cells. Using NAD+ as a substrate, PARP-1 repeatedly catalyzes the transfer of successive units of ADP-ribose moieties, via a unique 2′,1″-O-glycosidic ribose-ribose bond, to target proteins, finally producing the PAR chain [115]. With low levels of DNA damage, PARP-1 acts as a survival factor involved in DNA damage detection and repair. In contrast, with high levels of DNA damage, increased accumulation of PARP-1 promotes cell death. In the DNA base excision repair process, damaged bases are recognized by DNA glycosylase and are removed by cleavage of an N-glycosidic bond. Apurinic/apyrimidinic (AP) endonuclease cleaves the DNA backbone, thereby generating a single-strand DNA nick. PARP-1 recognizes the DNA nick as a single-strand break and facilitates poly ADP-ribosylation of target proteins. PARP-1 is subjected to self-assembly ADP-ribosylation (PAR) and binds to the X-ray repair cross-complementing 1 molecule (XRCC1) at the site of the single-strand DNA break [116]. XRCC1 interacts with DNA polymerase II (polB) and DNA ligase III that fills the DNA gap and completes the repair process, respectively (Figure 4, left panel). Consistent with the notion that PARP-1 plays an important role in mediating DNA repair, a majority of CR cells are dependent on PARP-1 and become susceptible to PARP inhibitor-induced apoptosis [117]. Recent reports showed that ovarian cancer cells that acquired CR expressed high levels of mitogen-activated protein kinase phosphatase-1 (MKP-1) and PARP-1 proteins; hence, silencing MKP-1 or PARP-1 increases cisplatin sensitivity in resistant cells. Moreover, the pharmacologic inhibition of PARP-1 activity restores cisplatin sensitivity in MKP-1 overexpressing cells [118]. PARP-1 knockout mice show increased mitochondrial numbers and oxygen consumption in a manner that mirrored SIRT1 activation [119]. Hence, removing PARP-1 as an enzyme that consumes NAD+ results in increased NAD+, SIRT1 activity, and OXME, thus linking the pathway involved in DNA repair to metabolic modulations in cells.

### 4.2. Cisplatin Resistance and SIRTs

The sirtuin family are NAD+-dependent histone deacetylases and ribosylases comprised of seven proteins denoted as SIRT1–7, which share a highly conserved NAD+-binding catalytic domain [120]. SIRTs are involved in diverse cellular processes including DNA repair, redox, and energy metabolism. Interestingly, it has been shown that increased SIRT1 can lead to an increase in the xCT pump and confer resistance to cisplatin in ovarian cancer cells [121]. In addition, SIRT1 overexpression significantly enhanced the resistance to cisplatin and paclitaxel in endometrial carcinoma cell lines, thus using the SIRT1 inhibitor (EX527) to suppress the proliferation of these cells [122]. SIRT2 can inhibit lipid synthesis by suppressing ATP citrate lyase (ACLY) [123]. ATP citrate lyase is the primary enzyme responsible for the synthesis of cytosolic acetyl-CoA (Figure 2). Studies have shown that downregulation of SIRT2 decreased the sensitivity to cisplatin treatment in NSCLC, but on the other hand, upregulating SIRT2 by resveratrol treatment sensitized NSCLC cells to cisplatin treatment [124]. As for SIRT3, it has been shown that it may associate with glutamine metabolism and augment the activity of GDH (glutamate dehydrogenase) [125]. In cultured human tubular cells, cisplatin reduced SIRT3 expression, resulting in mitochondrial fragmentation, but restoration of SIRT3 with AICAR (AMPK agonist) improved cisplatin-induced mitochondrial dysfunction. Thus, these data indicate that enhancing SIRT3 can improve mitochondrial dynamics as a potential strategy for improving outcomes of renal injury caused by cisplatin [126]. Unlike SIRT3, SIRT4 inhibited GDH and was shown to protect against the accumulation of DNA damage and reduced cell death in DNA damage caused by cisplatin [127,128]. Reports also indicated that SIRT4 reduced pyruvate dehydrogenase complex (PDH) activity (an enzyme converting pyruvate to acetyl CoA for TCA cycle; see Diagram1) [129]. SIRT5 is overexpressed in human NSCLC and served as a predictor of poor prognosis. SIRT5 regulates lung cancer resistance to cisplatin, 5-FU, and bleomycin in vitro and in vivo through regulating Nrf2 and its downstream genes [130]. SIRT6 promotes DNA repair under stress by activating PARP-1, also playing a critical role in glucose homeostasis [131]. In the case of gluconeogenesis, SIRT6 indirectly suppresses PGC-1α, leading to downregulation of hepatic glucose production [132]. Similar to SIRT1, SIRT6 can function as a repressor of the transcription factor Hif1α (a critical regulator of nutrient stress responses) and can shut down the glycolytic flux by deacetylation of histone H3 lysine 9 (H3K9) [133,134]. In contrast to SIRT3, increased SIRT7 levels result in increased expression of tumor necrosis factor-α (TNF-α), which in turn stimulates ROS production through NOX2 [135]. Furthermore, SIRT7 knockout mice were resistant to cisplatin-induced renal injury. These extensive findings demonstrate that alteration in the expression of the SIRT family of proteins is one of several modifications that contribute to cellular resistance to cisplatin and suggest that regulation of SIRTs might be an important target for therapy and might serve as a potential prognostic factor.

## 5. Cancer Cell and Immunometabolism

New forms of cancer therapy using the body’s own immune system to fight cancer have emerged lately. In particular, checkpoint inhibitor(s) such as nivolumab and pembrolizumab are FDA-approved agents. Moreover, increasing evidence suggests that downregulation of cellular metabolism plays a pivotal role in reducing the ability of the immune system to inhibit tumor growth. In the tumor microenvironment (TME), immune cells operate at a metabolic disadvantage since they are constrained by a lack of carbon nutrients due to the competition from the tumor cells [136]. In fact, we found that CR tumors undergo a “*second*” metabolic switch that utilized oxidative metabolism (OXMET) and increased amino acid uptake [6]. Consequently, this surrounding microenvironment is deprived of amino acids, creating an unfavorable condition for the viability of cytotoxic effector T cells (T eff), which are highly anabolic and require high amounts of amino acids for growth [137,138]. For instance, metabolite by-products of tryptophan catabolism such as kynurenine can inhibit T cell activation as well as cytolytic function and support immune suppressive regulatory T cell (T reg) differentiation [139]. HIF1α induced by TME hypoxia can also promote the generation and maintenance of T reg cells [140]. Moreover, HIF1α can also lead to the expression of programmed death ligand 1 (PD-L1) in myeloid-derived suppressor cells (MDSC), thereby mediating potent immunosuppressive functions in tumor-specific T eff cells [141]. Together, the metabolic and nutrient changes found in CR tumors can reshape TME and have a decisive role in immune functions.

### 5.1. CR Cells and PD-L1

Programmed death 1 (PD-1, CD279) and its ligand PD-L1 (CD274) are transmembrane proteins. PD-1 is predominantly expressed in activated T cells while PD-L1 is known to be expressed in many cancer cells. PD-1 and PD-L1 engagement dampens T effector cells function, which leads to impairment of effective immune response against the tumor; thus, overexpression of PD-L1 has been correlated with poor prognosis in NSCLC [142]. Importantly, reports indicated that cisplatin treatment can induce PD-L1 expression in various cancer types, including NSCLC [143]. The specific mechanisms of this activation pathway are still not clearly defined. Nevertheless, it has been shown that ERK and AKT signaling pathways can induce upregulation of PD-L1 in both antigen presenting cells and cancer cells through STAT3 [144]. STAT3 has been implicated in regulating the TME through several mechanisms, including the recruitment of myeloid-derived suppressor cells (MDSCs) or the decrease of immune cell infiltration in different types of tumors. When activated, STAT3 undergoes phosphorylation-induced homodimerization. The homodimer then translocates to the nucleus and binds to a PD-L1 promoter. STAT3 also activates DNA methyltransferase-1 (DNMT1), which methylates the promoter region and subsequently suppresses the expression of genes involved in immune surveillance, such as immunoproteasome subunits (PSM) B8 and B9 and the human leukocyte antigens (HLA) [145]. Consistent with this finding, it has been demonstrated that abrogation of DNMT1 can decrease PD-L1 expression and increase cisplatin sensitivity [146].

NSCLC possess a higher baseline of PD-L1, but a recent finding has shown that NSCLC is resistant to cisplatin and undergoes epithelial–mesenchymal transition to enable invasion/metastasis as well as to escape immune surveillance by expressing even higher PD-L1. Thus, it is possible that monotherapy with immune checkpoint inhibitors that do not demonstrate encouraging results as first-line therapy may be better used for CR cells instead. In addition, the relationship between STAT3 and PD-L1 expression in CR tumors also provides evidence that STAT3 can be used as a predictive response to immunotherapy, which warrants further investigation.

### 5.2. CR Cells and Kynurenine Pathway

The impact of tumor metabolism on the tumor microenvironment is not well understood. Given the fact that NAD+ is very important for cell survival, it is known that tryptophan (TRP) can be used for de novo biosynthesis of NAD+ through the kynurenine pathway to quinolinic acid. This is the universal metabolite in biology that generates the aromatic pyridine ring of NAD+ [147]. However, TRP is less efficient and a poor NAD+ precursor in vivo. TRP will only be diverted to the synthesis of NAD+ when its supply exceeds enzymatic capacity [148]. Thus, the main source of NAD+ comes from salvage pathways, which require the uptake of other NAD+ precursors, such as nicotinic acid (NA). In fact, NA is 60 times more efficient as a precursor of NAD+ when compared with TRP [149,150]. TRP is an essential amino acid, required for protein synthesis, and as the precursor of serotonin and melatonin, the catabolism of TRP is known to generate kynurenine (KYN) via the kynurenine pathway. Approximately 99% of ingested TRP not used for protein synthesis is catabolized by this pathway [151]. Importantly, indoleamine 2,3-dioxygenase (IDO; the rate-limiting step of the kynurenine pathway) can exploit superoxide as a substrate [152] and can in turn break down TRP, leading to the accumulation of KYN metabolites. Secretion of KYN to the tumor microenvironment plays a key role in reprogramming naïve T cells to the immune suppressive regulatory T cell (T reg) phenotype (Figure 5). Hence, these effects link tumor metabolism to the immune response localized to the tumor.

Interestingly, increased secretion of TRX1 from redox-stress and KYN levels in the TME can enhance regulatory T cell (T reg) infiltration and stimulate the conversion of naive T cells to T reg [153,154,155], creating an immunosuppressive environment. These factors in combination with challenging nutrient availability contribute to a unique microenvironment that favors the presence of T reg rather than cytotoxic T cells. Higher basal levels of KYN have been found in multidrug-resistant cells than in chemo-sensitive cells, and these higher levels have also displayed an inverse correlation with patient survival [156,157]. There are preclinical findings that IDO1 inhibitors may be able to safely empower the efficacy of cytotoxic or targeted chemotherapy, radiotherapy, and immune checkpoint therapy. However, the recent ECHO-301/KEYNOTE-252 phase III clinical trial (NCT02752074) tested the efficacy of IDO1 inhibitor with other chemotherapy in patients with advanced melanoma previously untreated with PD-1 or PD-L1 checkpoint inhibitors and did not yield very satisfactory results [158]. The study did not meet its primary objective of improvement in progression-free survival [159]. The reasons for this negative trial are undefined, but it is possible that patient selection maybe the culprit, suggesting that IDO1 inhibitors should be used in patients who have already failed chemotherapy such as cisplatin.

## 6. Concluding Remarks

Cisplatin therapy for the treatment of solid tumors has many different uses and multiple cellular targets that are still not well defined. The well-accepted mechanism of induced cell death is through the formation of DNA adducts and increased oxidative stress. The development of platinum resistance is inevitable and complex, involving many molecular pathways. Given the revival that tumor Cell Metab. has seen in recent years as a target for therapeutic intervention in cancer, there is a growing appreciation that tumors undergo a “*second*” metabolic switch when acquiring resistance to cisplatin. Alterations in metabolic pathways therefore create a window of selectivity for these resistant cells. A new paradigm establishing effective connections between tumor metabolism and tumor microenvironment is gaining traction but is still in a very early stage of elucidation. The major concern is that most metabolic and redox studies are primarily performed in cultured cancer cells, which may not accurately represent the metabolic and redox status of intact tumors or of immune cells.

We still have limited knowledge of how and to what extent metabolic rewiring causes drug resistance in solid tumors, but emerging data that we reviewed herein are promising and may lead to rational combinations of metabolic drugs with chemotherapeutic agents (Table 1) that could improve treatment outcomes. We also believe that targeting tumor metabolism and the proximal microenvironment in patients who fail cisplatin should lead to improvement of overall survival.

## Figures and Tables

**Figure 1 ijms-22-07379-f001:**
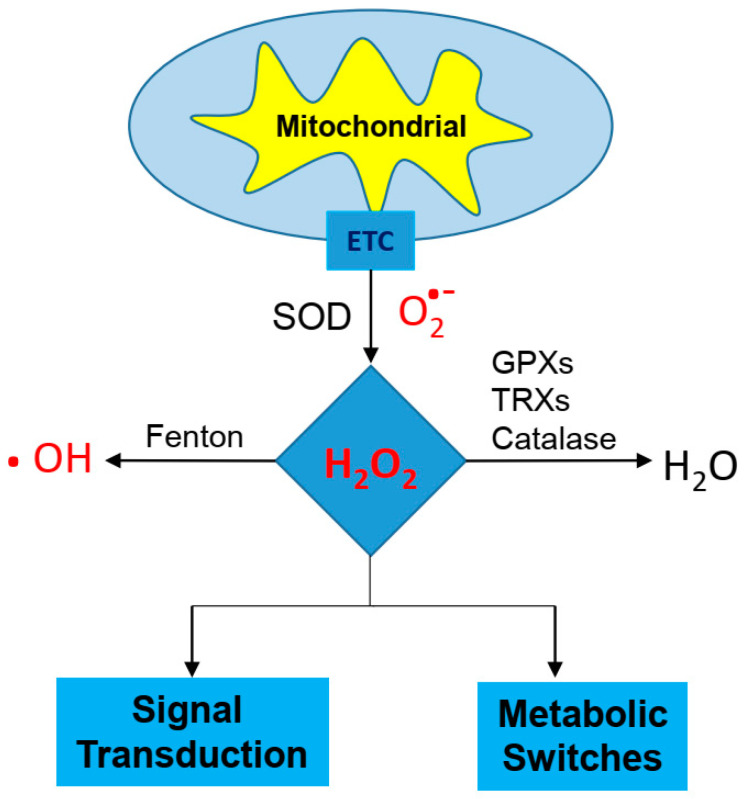
Electron leak generates from the electron transport chain of the mitochondria. Superoxide dismutase (SOD) enzymes convert superoxide molecules into a H_2_O_2_, which can then be reduced to water by glutathione peroxidases (GPXs), thioredoxins (TRXs), or catalase to water (H_2_O) molecules. Hydrogen peroxide can activate the signaling pathway and metabolic adaptations.

**Figure 2 ijms-22-07379-f002:**
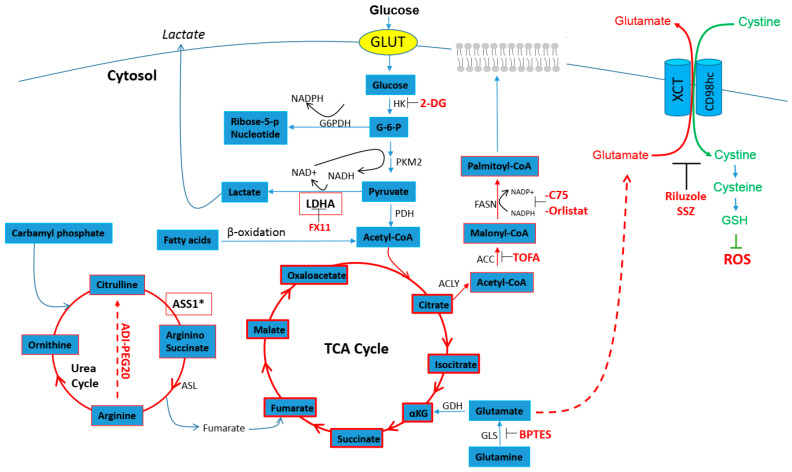
Metabolic scheme of bioenergetics and anabolic metabolism in cancer cells that represent treatment targets. CR cells take up large amounts of glutamine and use these nutrients to fuel the tricarboxylic acid (TCA) cycle and oxidative phosphorylation (OXPHOS). Glutamine is hydrolyzed to glutamate for glutathione synthesis, an essential factor to abrogate high ROS via xCT antiporter. Increased fatty acid synthesis enzymes are also found in CR tumors. Moreover, tumors lacking ASS1 enzyme in the urea cycle require extracellular arginine in the circulation for survival. ACLY = ATP citrate lyase, ACC = acetyl-CoA carboxylase, ADI-PEG = arginine deiminase pegylated-20, ASL = argininosuccinate lyase, ASS1 = argininosuccinate synthetase, BPTES = glutaminase inhibitor, CD98hc = CD98 heavy chain, 2-DG = 2-deoxy-d-glucose, FX= LDHA inhibitor, FASN = fatty acid synthase, G-6-P = glucose-6-phosphate, G6PDH = glucose-6-P dehydrogenase, GLS = glutaminase, GDH = glutamate dehydrogenase, GLUT = glucose transporter, GSH = glutathione, HK = hexokinase, α-KG = alpha-ketoneglutarate, LDHA = lactate dehydrogenase-A, PDH = pyruvate dehydrogenase, PKM2 = pyruvate kinase isozymes M2, PPP = pentose phosphate pathway. ROS = reactive oxygen species, SSZ = Sulfasalazine, TCA = tricarboxylic acid, XCT = system x_c_^-^ amino-acid transporter.

**Figure 3 ijms-22-07379-f003:**
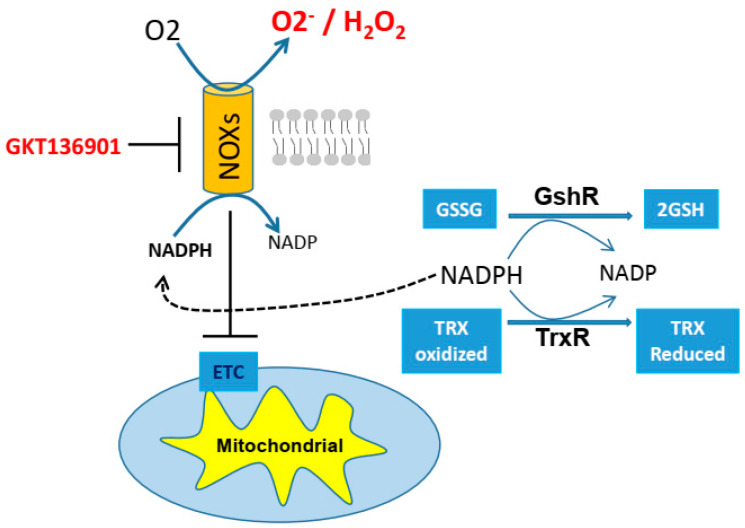
The dual role of NADPH. NADPH oxidase (NOX) can generate superoxide from oxygen and NADPH. On the other hand, NADPH is required for regeneration of GSH and TRX through glutathione reductase (GshR) and thioredoxin reductase (TrxR), respectively. ETC = electron transport chain, GSH = glutathione, GSSG = glutathione disulfide, NADPH = nicotinamide adenine dinucleotide phosphate, NADP = nicotinamide adenine dinucleotide phosphate, TRX = thioredoxin.

**Figure 4 ijms-22-07379-f004:**
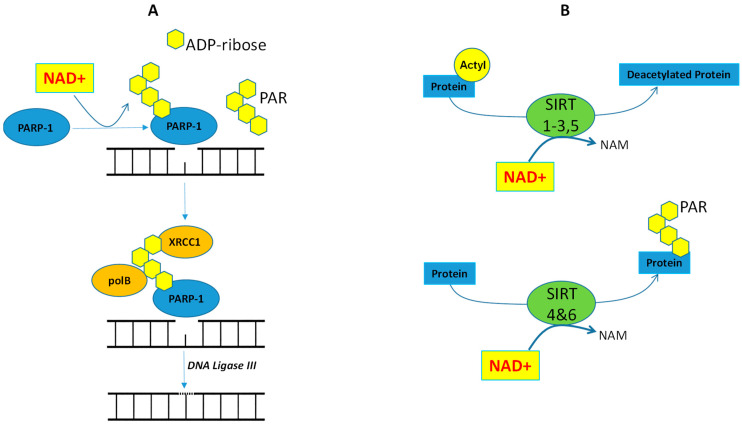
Panel (**A**) shows the role of PARP-1 in DNA base excision repair. PARP-1 binds to the damaged DNA and consumes nicotinamide adenine dinucleotide (NAD+) as a substrate to produce linear and branched polymers of ADP-ribose chain (PAR). PAR amplifies nucleosome remodeling by increasing the accessibility of base excision repair (BER) proteins such as X-ray repair cross-complementing protein 1 (XRCC1), DNA ligase III, and DNA polymerase β (pol β) to the damaged DNA. Panel (**B**) indicates two reactions catalyzed by sirtuins: (i) deacetylation and (ii) ADP-ribosylation. SIRT1–SIRT3 and SIRT5 catalyze a deacetylation reaction in which an acetyl group is transferred to the ADP-ribose (ADPR) moiety of NAD+. In contrast, SIRT4 and SIRT6 catalyze ADP-ribosylation of proteins rather than deacetylation. ADP = Adenosine di-phosphate, NAM = nicotinamide, PARP = poly (ADP-ribose) polymerase 1, SIRT = sirtuin.

**Figure 5 ijms-22-07379-f005:**
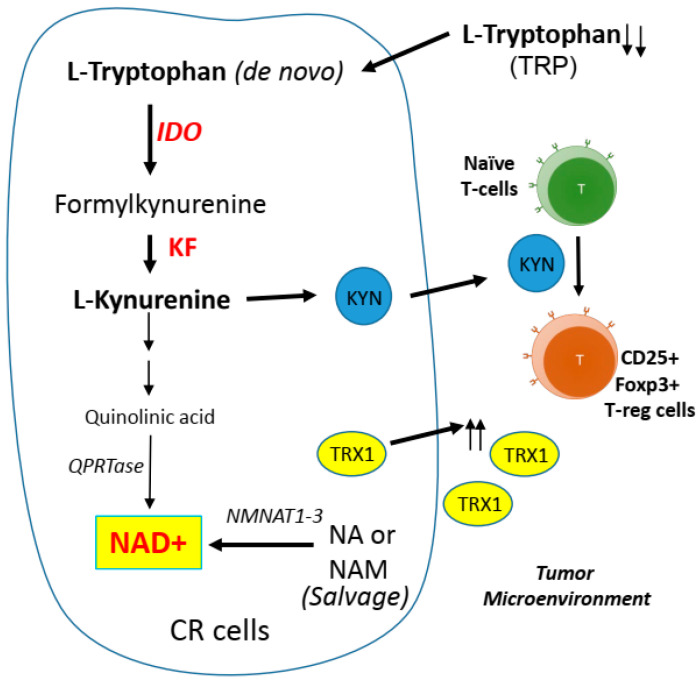
Schematic of tryptophan catabolism via the kynurenine pathway. Indoleamine 2,3-dioxygenase (IDO) plays a key role in the regulation of the immune system. Increased kynurenine (KYN) and thioredoxin-1 (TRX1) secretion from cisplatin resistant (CR) cells further enhances expansion of the regulatory T cell (T reg) population. KF = kynurenine formamidase, NA= nicotinic acid, NAM= nicotinamide (NAM), NMNAT = nicotinamide mononucleotide adenylyltransferase 1–3, QPRTase = quinolinate phosphoribosyltransferase.

**Table 1 ijms-22-07379-t001:** Example of potential metabolic targets for cancer therapy.

Drug Name	Target (Propose Target)	References
Glycolysis pathway		
2-DG	HK	[160,161]
FX-11	LDHA	[6]
Oxamate	LDHA	[162]
DCA	PDK	[163]
Riluzole	LDHA via NAD+	[6,164]
Glutaminolysis pathway		
BPTES	GLS	[165]
CB-839	GLS	[166,167]
EGCG	GDH	[168,169]
Fatty acid synthesis pathway		
C75	FASN	[22,170,171]
Cerulenin	FASN	[171]
Orlistat	FASN	[171,172]
TOFA	ACC	[22,171]
Arginine synthesis pathway		
ADI-PEG20	arginine degradation	[54,173,174]
Arginase1	arginine degradation	[51,175]
Redox pathway		
BSO	GSH	[100,176,177]
PX-12	TRX	[178]
Elesclomol	ETC (Cu++)	[22,88,179]
Riluzole	xCT	[6,164]
SSZ	xCT	[43,180]
EX527	SIRT1	[122]
Kynurenine pathway		
BCH	LAT1	[173,174]
CH-223191	AHR	[181]
Epacadostat	IDO	[176]
Indoximod	IDO	[176]
Navoximod	IDO	[176]

## Data Availability

All data are provided in the manuscript.

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
