# Peer review of "Cisplatin Resistance and Redox-Metabolic Vulnerability: A Second Alteration"

_ijms, 2021, doi:10.3390/ijms22147379_

Round 1
Reviewer 1 Report
The authors in their article present an overview of research related to the mechanism of cisplatin acquired resistance. The main focus of this manuscript is the second metabolic switch from glycolysis to oxidative metabolism that cells undergo in the process of cisplatin resistance development. Overall, the manuscript is well-written and well-presented. It is a comprehensive overview of redox-metabolism changes that mediate cisplatin resistance. I have the following comments for the authors:
In Table 1 the authors present a general overview of metabolic targets that could potentially increase efficacy of cisplatin treatment. Since this manuscript is related to cisplatin resistance, the authors could define in this table the drugs and targets that have been shown to increase cisplatin efficacy. Furthermore, the authors could elaborate, which strategies have improved cisplatin activity, as this information could be interesting to the reader of this article.
Although the manuscript is well-written, please recheck the manuscript for any language or editing errors. Some corrections are provided below:
Line 58: high proliferation rate and growth
Line 79: resist a cytotoxic drug
Line 172: can selectively kill CR cells in vitro
Line 291: on NAD+.
Line 306: in turn generate superoxide.
Line 346: It has been known that the decrease of NAD+ enhanced PARP activity correlates with a downregulation of SIRT activity – please rephrase
Line389, 402: the authors refer to Diagram1, not present in the manuscript
Author Response
We would like to thank the reviewers for their constructive comments and suggestions and for their efforts to improve this review. We have revised based on your input. Responses to the critiques are summarized below along with changes that have been introduced into the text of the manuscript.
Responses to Reviewer 1
-We have changed the reference in Table 1 to reflect metabolic targets that could potentially increase efficacy of cisplatin treatment. Moreover, there are many published strategies to improve cisplatin resistance including blocking transport (either facilitate influx or block efflux) or inhibit DNA repair, etc. However, it often affect normal cells which hamper their use in the clinic. We have combined ADI-PEG20 (arginine degrading enzyme) with cisplatin before to treat ASS1 (argininosuccinate synthetase) deficiency tumor in order to inhibit DNA repair with some success, but ADI-PEG20 has not yet approved by FDA.
-We have made corrections to language and editing errors.
Reviewer 2 Report
I have appreciated the reading of the present review. It is well written and clear in its meaning,
In my opinion is suitable to be published in the journal.
Some minor concers:
- the abstract is a little bit confusing (platinum-based chemotherapy). Please, specify better
- please, avoid the repetition and the abbreviations (only the first time for each term is correct the use of the abbreviation).
- in my opinion Table 1 is not put at the right point of the text
Author Response
We would like to thank the reviewers for their constructive comments and suggestions and for their efforts to improve this review. We have revised based on your input. Responses to the critiques are summarized below along with changes that have been introduced into the text of the manuscript.
Reponses to Reviewer 2
-We have rewritten the abstract to make it clearer.
-We have edited the manuscript and remove repetitive abbreviation.
Round 2
Reviewer 1 Report
The authors have made the appropriate corrections.